# Methylglyoxal and D-lactate in cisplatin-induced acute kidney injury: Investigation of the potential mechanism via fluorogenic derivatization liquid chromatography-tandem mass spectrometry (FD-LC-MS/MS) proteomic analysis

**Shih-Ming Chen[1], Tsung-Hui Chen[1], Hui-Ting Chang[1,2], Tzu-Yao Lin[1], Chia-Yu Lin[1,3], Pei-Yun Tsai[1,4], Kazuhiro Imai[5], Chien-Ming Chen[6], Jen-Ai Lee⊙[1]***

1 Department of Pharmacy, School of Pharmacy, Taipei Medical University, Taipei, Taiwan, 2 Department of Health, Taipei City Government, Taipei, Taiwan, 3 Department of Pharmacy, Taipei Medical University Hospital, Taipei, Taiwan, 4 Department of Pharmacy, Wan-Fang Hospital, Taipei, Taiwan, 5 Research Institute of Pharmaceutical Sciences, Musashino University, Tokyo, Japan, 6 Department of Electro-Optical Engineering, National Taipei University of Technology, Taipei, Taiwan

* jenai@tmu.edu.tw

## Abstract

Nephrotoxicity severely limits the chemotherapeutic efficacy of cisplatin (CDDP). Oxidative stress is associated with CDDP-induced acute kidney injury (AKI). Methylglyoxal (MG) forms advanced glycation end products that elevate oxidative stress. We aimed to explore the role of MG and its metabolite D-lactate and identify the proteins involved in CDDP-induced AKI. Six-week-old female BALB/c mice were intraperitoneally administered CDDP (5 mg/kg/day) for 3 or 5 days. Blood urea nitrogen ($42.6 \pm 7.4$ vs. $18.3 \pm 2.5$; $p < 0.05$) and urinary $N$-acetyl-β-D-glucosaminide (NAG; $4.89 \pm 0.61$ vs. $2.43 \pm 0.31$ U/L; $p < 0.05$) were significantly elevated in the CDDP 5-day group compared to control mice. Histological analysis confirmed AKI was successfully induced. Confocal microscopy revealed TNF-α was significantly increased in the CDDP 5-day group. Fluorogenic derivatized liquid chromatography-tandem mass spectrometry (FD-LC-MS/MS) showed the kidney MG ($36.25 \pm 1.68$ vs. $18.95 \pm 2.24$ mg/g protein, $p < 0.05$) and D-lactate ($1.78 \pm 0.29$ vs. $1.12 \pm 0.06$ mol/g protein, $p < 0.05$) contents were significantly higher in the CDDP 5-day group than control group. FD-LC-MS/MS proteomics identified 33 and nine altered peaks in the CDDP 3-day group and CDDP 5-day group (vs. control group); of the 35 proteins identified using the MOSCOT database, 11 were antioxidant-related. Western blotting confirmed that superoxide dismutase 1 (SOD-1) and parkinson disease protein 7 (DJ-1) are upregulated and may participate with MG in CDDP-induced AKI. This study demonstrates TNF-α, MG, SOD-1 and DJ-1 play crucial roles in CDDP-induced AKI.

**Data Availability Statement:** All relevant data are within the paper and its Supporting Information files.

**Funding:** Cathay General Hospital supported the financial funding (108CGH-TMU-06) to SMC. We are grateful to the Yung Shin Pharmaceutical Co. (Taiwan) for providing an API 4000 triple quadrupole mass spectrometer. The funders had no role in study design, data collection and analysis, decision to publish, or preparation of the manuscript.

**Competing interests:** We are grateful to the Yung Shin Pharmaceutical Co. (Taiwan) for providing an API 4000 triple quadrupole mass spectrometer. This does not alter our adherence to PLOS ONE policies on sharing data and materials.

# Introduction

Cisplatin (*cis*-diamminedichloroplatinum II; CDDP) is widely used as a chemotherapeutic agent for a variety of cancers. However, one-third of patients treated with CDDP develop nephrotoxicity, which limits the application of this drug [1, 2]. CDDP-induced nephrotoxicity is associated with damage to the S1 and S3 segments of the proximal tubules. Moreover, high concentrations of CDDP induce necrosis of the kidney tissues, whereas lower concentrations can promote apoptosis in human kidney cells [3].

CDDP causes reactive oxygen species (ROS)-induced oxidative stress and lipid peroxidation, which result in membrane dysfunction and production of toxic metabolites. A relationship between tumor necrosis factor-α (TNF-α) and CDDP-induced nephrotoxicity was previously described [4–7].

Cisplatin-induced nephrotoxicity is mediated by oxidative stress [1, 2, 8]. The balance between ROS and antioxidant enzymes affects the degree of organ damage [9]. Methylglyoxal (MG), also known as 2-oxopropanal or pyruvaldehyde, contains two carbonyl groups and reacts with proteins and nucleic acids to generate MG-adducts. These advanced glycation end products can lead to protein denaturation and malfunctions that induce mitochondrial dysfunction and cellular apoptosis [10–13]. The levels of MG and its metabolite D-lactate correlate with oxidative stress in animal models of severe kidney injury [14–23]. However, the precise involvement of MG and D-lactate in CDDP-induced nephrotoxicity are unclear.

Proteomics, the study of proteins in biological specimens and the interactions between proteins and pathological conditions [24–26], has provided significant insight into various mechanisms of disease [27]. Two-dimensional polyacrylamide gel electrophoresis (2D-PAGE) is widely applied in proteomic studies, but is a complicated technique with low sensitivity and reproducibility [28]. Fluorogenic derivatization-liquid chromatography-tandem mass spectrometry (FD-LC-MS/MS) can successfully identify proteins in biological samples. A series of studies applied FD-LC-MS/MS to screen unknown proteins and identify their roles under physiological and pathological conditions in animal tissue and human cell lines [26, 29, 30].

In this study, we aimed to confirm whether MG and D-lactate play a role in CDDP-induced kidney injury. Specifically, a FD-LC-MS/MS proteomic method was employed to explore the proteins involved in a mouse model of oxidative stress-related acute kidney injury (AKI) induced by CDDP.

# Materials and methods

## Chemicals

Bovine serum albumin (BSA), 4-methylumbelliferyl *N*-acetyl-β-D-glucosaminide (4-MU-NAG), 4-methylumbelliferone (4-MU), the periodic acid-Schiff (PAS) kit, citric acid, ammonium chloride ($NH_4Cl$), propionic acid, tris(2-carboxethyl)phosphine hydrochloride (TCEP), guanidine buffer, calcium chloride ($CaCl_2$), ammonium bicarbonate ($NH_4HCO_3$), sodium dodecyl sulfate (SDS) and 30% acrylamide were purchased from Sigma Chemical Co. Ltd (St Louis, MO, USA); 5,6-diamino-2,4-hydroxypyrimidine sulfate (DDP), 4-nitro-7-piperazino-2,1,3-benzoxadiazole (NBD-PZ), 2,2-dipyridyl disulfate (DPDS), triphenyl phosphine (TPP) and 7-chloro-N-[2-(dimethylamino) ethyl]-2,1,3-benzoxadiazole-4-sulfonamide (DAABD-Cl) were purchased from Tokyo Kasei Chemicals (Tokyo, Japan); 3-[(3-cholamido-propyl) dimethylammonio] propanesulfonic acid (CHAPS), trifluoroacetic acid (TFA) and ethylenediaminetetraacetic acid disodium salt (EDTA•2Na) were purchased from Wako Pure Chemicals Industries Ltd (Tokyo, Japan). HPLC-grade acetonitrile (ACN), methanol (MeOH) and MS-grade formic acid (FA) were obtained from Merck (Darmstadt, Germany).

Derivatives were separated and collected using high-performance liquid chromatography with a fluorescence detector (FD-HPLC; Hitachi, Tokyo, Japan). After digesting the isolated derivatives, the samples were identified by LC-MS/MS (4000 QTRAP; Applied Biosystems, Foster City, CA, USA) with the MASCOT database searching system. Trypsin was purchased from Promega (Fitchburg, WI, USA).

## Animal model of CDDP-induced AKI

All animal protocols followed the ethical guidelines of the Institutional Animal Care and Use Committee of Taipei Medical University (LAC-2019-0167). Twenty-five six-week-old female BALB/c mice were purchased from the National Laboratory Animal Center Foundation and randomly allocated to three groups, control, CDDP administration for 3 days (CDDP 3 days) and CDDP administration for 5 days (CDDP 5 days). Five animals were allocated into one group. All animals were allowed to acclimatize to the environment for a week before the experiments started. All animals had free access to water and food and were housed at 21 ˚C and 70–80% relative humidity under a normal feeding environment.

To induce acute kidney injury (AKI), the mice were injected with CDDP (5 mg/kg/day) intraperitoneally for 3 or 5 days. Mice were humanely sacrificed on day 4 or day 6 and the kidneys were dissected and stored at -80 ˚C directly for further analysis [14, 26]. Blood samples were obtained when the mice were sacrificed. Animals were sacrificed under isoflurane anesthesia, and all efforts were made to minimize suffering.

## Biochemical assays

A specialized metabolism device (Tokiwa Precious Metals, Japan) was used to collect 12-hour urine samples on day 3 and day 5 [14, 26].

Renal function was evaluated by assessing blood urea nitrogen (BUN), *N*-acetyl-β-D-glucosaminide (NAG) and urinary creatinine. BUN and urinary creatinine were determined using a dimension clinical chemistry system (Dade Behring Inc., Deerfield, IL, USA).

NAG activity was measured using a fluorometric assay to detect 4-MU [14, 26]. In brief, 4-MU-NAG was reacted with NAG in the urinary samples in 100 mM citrate buffer (pH 4.6–5.0) at 37 ˚C. The reaction was terminated after 15 min by the addition of 200 mM glycine buffer (pH 10.4–10.6), and the fluorescence intensity of 4-MU was quantified at excitation/emission wavelengths of 370/460 nm.

## Histopathologic analysis

Approximately half of the left kidney of each mouse was stained with periodic acid–Schiff (PAS) reagent following a previously published method [31]. Sections were observed at 200× using a light microscope.

The degree of kidney injury was assessed using the tubulointerstitial histological score (TIHS), which assesses degeneration of the tubular epithelium (scored from 0–5), mononuclear cell infiltration into the interstitium (scored from 0–3), and interstitial fibrosis (scored from 0–5). A higher TIHS score represents more severe damage [26, 31, 32].

## Immunofluorescence assay

Approximately half of the left kidney of each mouse was subjected to immunofluorescence analysis. Frozen kidney sections were rinsed in phosphate buffered saline (PBS) at room temperature, blocked in 10% control rabbit serum and incubated with TNF-α primary antibody (1:200); sc-52746; Santa Cruz Inc., Dallas, TX, USA). TRITC-labeled IgG (Santa Cruz Inc.,

USA) was used as a secondary antibody, then the samples were washed three times in PBS, mounted and the fluorescent signals were detected using an Olympus Fluoview FV500 Laser Scanning Confocal System (Tokyo, Japan).

## Preparation of homogenized kidney samples

Approximately 500 mg of tissue from the right kidney was homogenized in 500 μL of PBS using a Precellys 24 Tissue Homogenizer (Brevet Bertin Technologies, Montignyle, France) at 6000 rpm for 30 sec, centrifuged at 8000 rpm for 20 min at 4 ˚C, the supernatant was collected and the total protein content was quantified using the Pierce™ BCA Protein Assay Kit (Thermo Fisher Scientific Inc, Waltham, MA, USA) using BSA as a protein standard.

## Analysis of MG in homogenized kidney samples

The kidney MG content was determined by FD-HPLC according to previously published methods [14, 21, 33]. Briefly, 20 μL homogenized kidney samples were incubated with DDP at 60 ˚C for 30 min in 0.5 M ammonium chloride buffer (pH 10.0). The reaction was stopped by addition of 0.01 M citric acid (pH 6.0). Samples (20 μL) were injected onto the HPLC and separated using an ODS column (150 × 4.6 mm, 5 μm particle size; Biosil Chemical Co. Ltd., Taipei, Taiwan). The mobile phase was 97:3 (*v/v*) 0.01 M citric acid buffer (pH 6.0)/ACN and the flow rate was 0.7 mL/min. The temperature was fixed at 33 ˚C. The detector was set to excitation/emission wavelengths of 330/500 nm. Quantification was performed by integrating the height of the corresponding peaks on the chromatograms using a D-7500 integrator (Hitachi) and adjusting to the protein concentration of the tissue sample.

## Analysis of D-lactate in homogenized kidney samples

A two-dimensional column-switching FD-HPLC system was used to determine kidney D-lactate content [22, 32]. Homogenized kidney samples (20 μL) were mixed with 10 μL of propionic acid (as an internal standard) and 170 μL ACN and centrifuged at 700 *g* for 10 min at 4 ˚C to precipitate protein. Then, 100 μL of the supernatant was derivatized by adding 100 μL of 8 mM NBD-PZ, 25 μL of 280 mM DPDS and 25 μL of 280 mM TPP. The reactions were incubated at 30 ˚C for 3 h and stopped by addition of 250 μL of 0.1% TFA $_{(a.q.)}$. The samples were purified using MonoSpin™ SCX cartridges (GL Science Inc., Tokyo, Japan) to remove excess derivatizing agent.

The lactate derivatives were first separated on an Aqu-ODS-W-5u column (250 × 4.6 mm, 5 μm particle size; Biosil Chemical Co. Ltd, Taipei, Taiwan) using 68:12:20 (*v/v/v*) $H_2O$/ACN/ MeOH as the mobile phase at 30 ˚C. The flow rate was 0.7 mL/min for 0–35 min and 0.9 mL/ min for 35.1–60 min. The fraction containing the lactate derivatives was collected and injected onto a Chiralpak AD-RH column (150 × 4.6 mm, 5 μm particle size; Daicel Co. Osaka, Japan) to separate D-lactate and L-lactate. The mobile phase was 40:60 (*v/v*) $H_2O$/ACN at a flow rate of 0.3 mL/min. The detector for detecting the fraction containing lactate derivatives was set to excitation/emission wavelengths of 330/500 nm. The detectors for detecting D/L-lactate derivatives was set to excitation/emission wavelengths of 491/547 nm. Quantification was performed by integrating the areas of the corresponding peaks on the chromatograms (D-7500 integrator; Hitachi) and adjusting to the protein concentration of each sample.

## Sample preparation and protein derivatization

Approximately 500 mg of tissue from the kidney was homogenized in 2000 μL of 10 mM CHARS using a Precellys 24 Tissue Homogenizer (Brevet Bertin Technologies, Montignyle,

**Table 1. Gradient elution program for FD-HPLC proteomic analysis.**

| Phase | Time (min) | | | | | | | | | | | | | | |
|---|---|---|---|---|---|---|---|---|---|---|---|---|---|---|---|
| | 0 | 10 | 15 | 40 | 60 | 90 | 140 | 150 | 200 | 205 | 420 | 500 | 530 | 560 | 570 |
| A (%) | 94 | 94 | 35 | 30 | 27 | 1 | 1 | 0 | 0 | 0 | 0 | 0 | 0 | 0 | 0 |
| B (%) | 5 | 5 | 30 | 30 | 38 | 44 | 44 | 47 | 48 | 51 | 60 | 70 | 90 | 90 | 100 |
| C (%) | 1 | 1 | 35 | 35 | 35 | 55 | 55 | 53 | 52 | 49 | 40 | 30 | 10 | 10 | 0 |

France) at 6000 rpm for 30 sec, centrifuged at 8000 rpm for 20 min at 4 ˚C. The supernatant was collected and the total protein content was quantified by using the pervious method. Tissue homogenates were diluted into 4.0 mg/mL protein by 10 mM CHAPS. 20 μL of 50 mM CHAPS, 20 μL of 2.5 mM TCEP, 20 μL of 10 mM EDTA·2Na, 25 μL of 8 M guanidine and 5 μL of 140 mM DAABD-Cl were added to 10 μL of the diluted samples. The reactions were incubated at 40 ˚C for 10 min and stopped by adding 20% TFA.

## FD-HPLC proteomic analysis

The FD-HPLC conditions were modified from a previous method [24–26]. The system was composed of a Hitachi HPLC system and L-2485 fluorescence detector at excitation/emission wavelengths of 395/505 nm. Derivatized protein samples (40 μL) were injected into the HPLC system at a flow rate of 0.55 mL/min. Mobile phase A was 9:1:90:0.15 (*v/v/v/v*) ACN/isopropanol/$H_2O$/TFA, mobile phase B was 69:1:30:0.15 ACN/isopropanol/$H_2O$/TFA (*v/v/v/v*) and mobile phase C was 4:1:95:0.20 (*v/v/v/v*) ACN/isopropanol/$H_2O$/TFA. The gradient elution program is shown in Table 1. A reverse-phase column (Intrada WP-RP, 250 x 4.6 mm i.d., Imtakt Co., Kyoto, Japan) was used to separate proteins at 60 ˚C based on polarity. The peaks obtained from the control and CDDP-treated mouse kidney samples were compared and the differential peaks were isolated and identified using LC-MS/MS and the MASCOT database [24–26].

## Protein identification

The organic mobile phase was evaporated using a Savant Speed Vac (Model SPD111V; Thermo Fisher Scientific Inc, Waltham, MA, USA). Then, 2.5 μL of 10 mM $CaCl_2$, 20 μL of 50 mM $NH_4CO_3$ and 2.5 μL trypsin were added to the protein residues and incubated at 37 ˚C for 2 h. The samples were loaded onto a nanopore-column (Zorbax 300SB-C18; 5 x 0.3 mm I. D.; Agilent, Santa Clara, CA, USA) and eluted using 0.10% TFA in 2.0% ACN at a flow rate of 10 μL/min. The mixtures were then separated on a $C_{18}$ nanoflow column (75 μm i.d. x 150 cm, C18 NanoEase, particle size 3.5 μm) at a flow rate of 0.5 μL/min. Mobile phase D was 2:98:0.1 (*v/v/v*) ACN/$H_2O$/FA and mobile phase E was 98:2:0.1 (*v/v/v*) ACN/$H_2O$/FA. The gradient elution program is shown in Table 2.

Peptide samples were injected into a API 4000 QTRAP MASS spectrometer (Agilent, Santa Clara, CA, USA) via a distal coated fused-silica needle (75 μm tube i.d., 15 μm tip i.d.; PicoTip™ Emitter, New Objective, MA, USA). One-second MS/MS scans were conducted for each

**Table 2. Gradient elution program for LC-MS/MS for proteomic analysis.**

| Phase | Time (min) | | | | | |
|---|---|---|---|---|---|---|
| | 0 | 5 | 32 | 35 | 38 | 45 |
| D (%) | 95 | 95 | 60 | 20 | 20 | 95 |
| E (%) | 5 | 5 | 40 | 80 | 80 | 5 |

precursor ion. Ions with a *m/z* ratio between 350 and 1250 were fragmented using capillary energies ranging from 1300–2500 V; the temperature of the interface heater was 150 ˚C.

After detection, data was submitted to MASCOT, a protein identification program, which is one of the LC-MS/MS protein identification system by Matrix Science Ltd. (www. matrixscience.com/). MASCOT is widely-used protein identification search engine. The proteins were identified using MASCOT version 2.2, NCBInr as the database, 0.5 Da as peptide tolerance and MS/MS tolerance, 2+ for the peptide charge. As previous research described [25–27, 30, 31].

### Western blotting

Homogenized kidney samples (15 μg per lane) were separated on 12% SDS/polyacrylamide gels and electrophoretically transferred onto PVDF membranes. The membranes were incubated with primary antibodies against mice parkinson disease protein 7 (DJ-1; 1:5000 dilution; GTX132552, GeneTex, CA, USA), superoxide dismutase 1 (SOD1; 1:3000 dilution; 100554; GeneTex) and β-actin (rabbit polyclonal beta actin antibody (1:2000 dilution; 20536–1-AP; Proteintech, Rosemont, IL, USA) as an internal control. HRP-conjugated Affinipure Goat Anti-Mouse IgG (H+L) (1:4000; SA00001–1; Proteintech) was used as the secondary antibody. The signals were determined using the enhanced chemiluminescence kit (BIOTOOLS, New Taipei City, Taiwan). The relative levels of DJ-1 and SOD-1 were assessed semi-quantitatively, normalized to β-actin and expressed relative to the respective controls using Image J (National Institute of Mental Health, Bethesda, MD, USA).

## Results

### Biochemical analysis of CDDP-induced AKI

The biochemical analysis is summarized in Fig 1 Compared to control mice at the same time points, urinary creatinine was significantly lower ($p < 0.05$) at days 3 and 5 in the mice injected with CDDP. Moreover, BUN was significantly higher at day 5 (but not day 3) and NAG was significantly higher at days 3 and 5 in the CDDP group compared to the control group. These biochemical data indicate that administration of CDDP for 3 and 5 days damaged the kidney function of the mice.

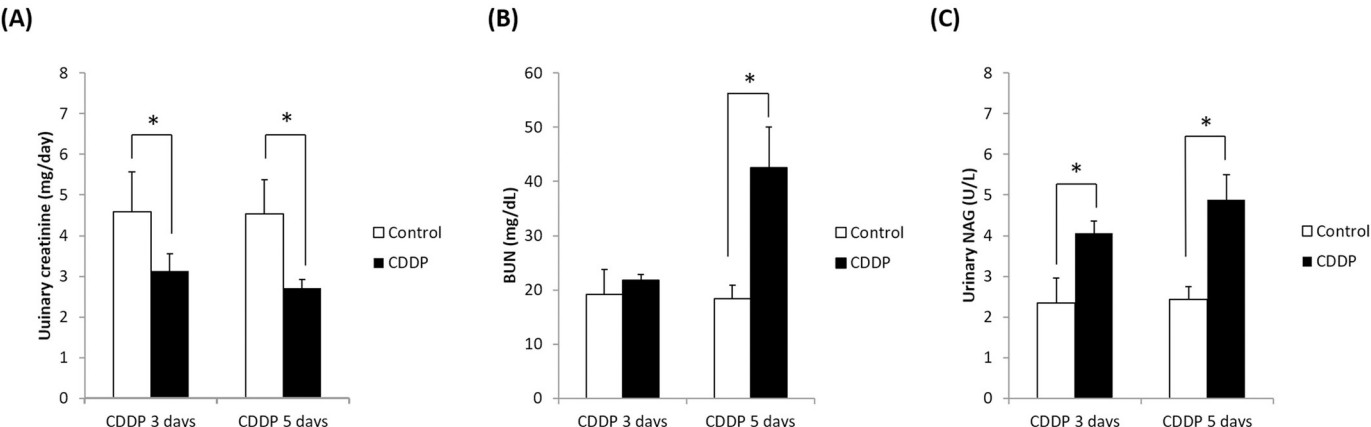

**Fig 1. Biochemical analysis of CDDP-induced AKI.** (A) Urinary creatinine, (B) NAG activity and (C) BUN in the control group, CDDP 3-day group and CDDP 5-day group. NAG: *N*-acetyl-acetylglucosaminidase, BUN: blood urea nitrogen, *p < 0.05, Student's T-test. N = 5 in each group, every experiment was conducted in triplicate.

## PAS staining and confocal imaging

Compared to the control group, the mice administered CDDP for 3 and 5 days exhibited damage to the renal tubules, including tubular cell atrophy and cell infiltration. The TIHS of the PAS-stained tissues confirmed that CDDP-induced pathological changes were observed in both the CDDP 3-day group and CDDP 5-day group. The damage was more severe in the CDDP 5-day group than the CDDP 3-day group (Fig 2).

Confocal microscope images of immunofluorescently stained kidney sections are shown in Fig 3. TNF-α was expressed at low levels in the control group and the highest levels in the CDDP 5-day group. Overall the results if biochemical, histological and Immunofluorescence assay indicated that CDDP induced kidney damage.

## MG content of kidney tissues

We used reverse-phase HPLC coupled to a fluorescence detector to quantify the MG contents of the mice kidney tissues (Fig 4; S1 Fig). The MG contents were 0.019± 0.002 for the control

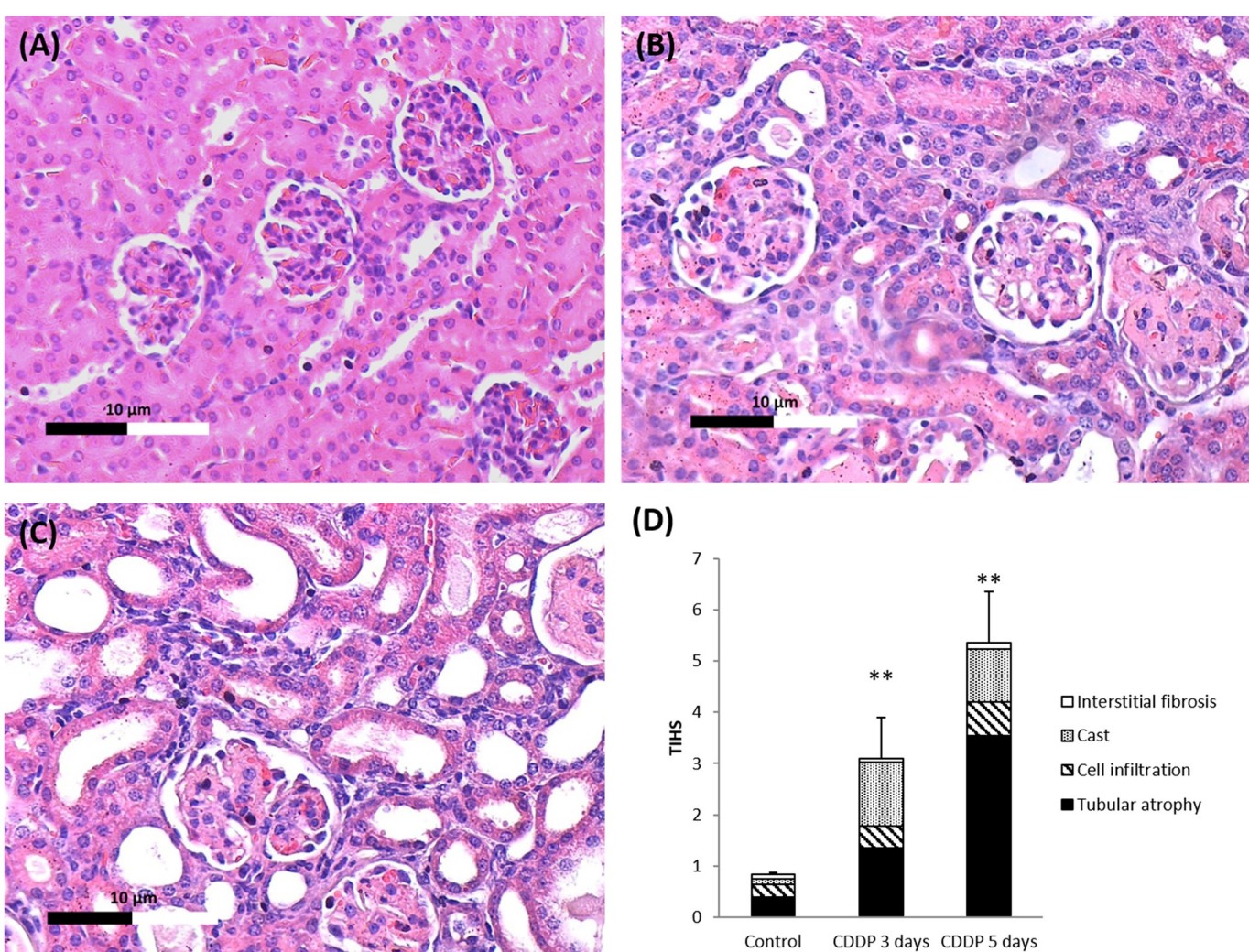

**Fig 2. Periodic acid Schiff (PAS)-stained kidney tissues.** (A) control group (B) CDDP 3-day group and (C) CDDP 5-day group (x200). Both the CDDP 3-day group and CDDP 5-day group exhibited renal tubule damage, such as tubular cell atrophy and cell infiltration. (D) Tubulointerstitial histological scores (TIHS) of the PAS-stained tissues; **p < 0.01 vs. control group, Student's T-test. N = 5 in each group, every experiment was conducted in triplicate.

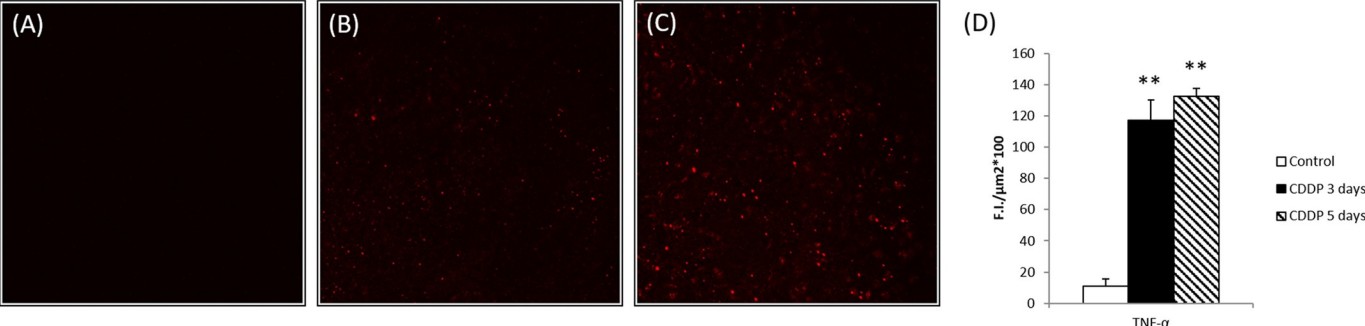

**Fig 3. Confocal microscopy of TNF-α expression in the kidney tissue.** (A) control group (B) CDDP 3-day group (C) CDDP 5-day group (D) Semi-quantitative analysis of the levels of TNF-α based on the confocal microscopy images; **p < 0.01 vs. control group, Student's T-test. N = 5 in each group, every experiment was conducted in triplicate.

group, 0.022 ± 0.002 for the CDDP 3-day group and 0.036 ± 0.002 μg/g protein for the CDDP 5-day group. The MG content of the CDDP 5-day group, but not the CDDP 3-day group, was significantly higher ($p < 0.05$) than the MG content of the control group. This result confirms that administration of CDDP for 5 days significantly increased the production of MG in the kidney.

## D-Lactate content in kidney tissues

A two-dimensional column switching system with a fluorescence detector was to quantify the D-lactate contents of the kidney tissues (Fig 5; S2 Fig). The D-lactate content was 1.12 ± 0.06 in the control group, 1.39 ± 0.35 in the CDDP 3-day group and 1.78 ± 0.29 nmol/mg protein in the CDDP 5-day group. Although there was no significant difference between groups, the D-lactate content tended to increase as the MG content increased.

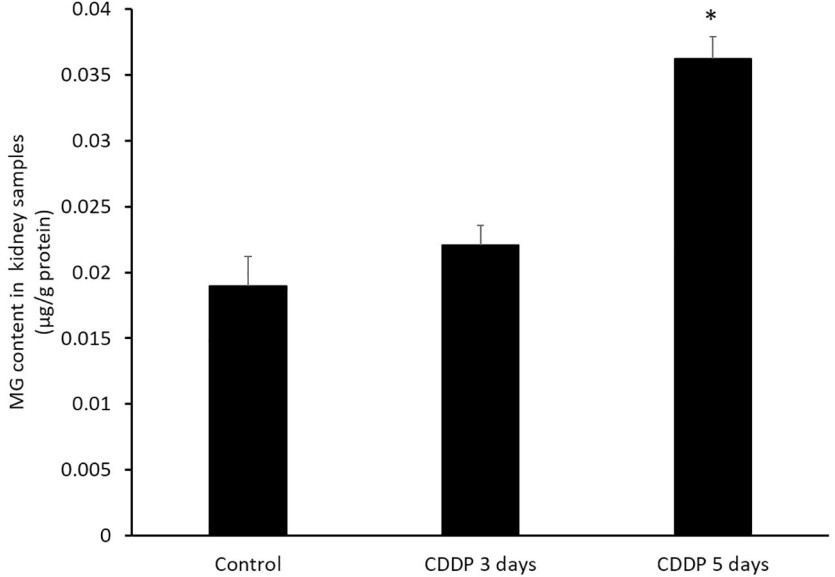

**Fig 4. The quantification results of methylglyoxal content in the kidney tissues.** *p < 0.05 vs. control group, Student's T-test. N = 5 in each group, every experiment was conducted in triplicate.

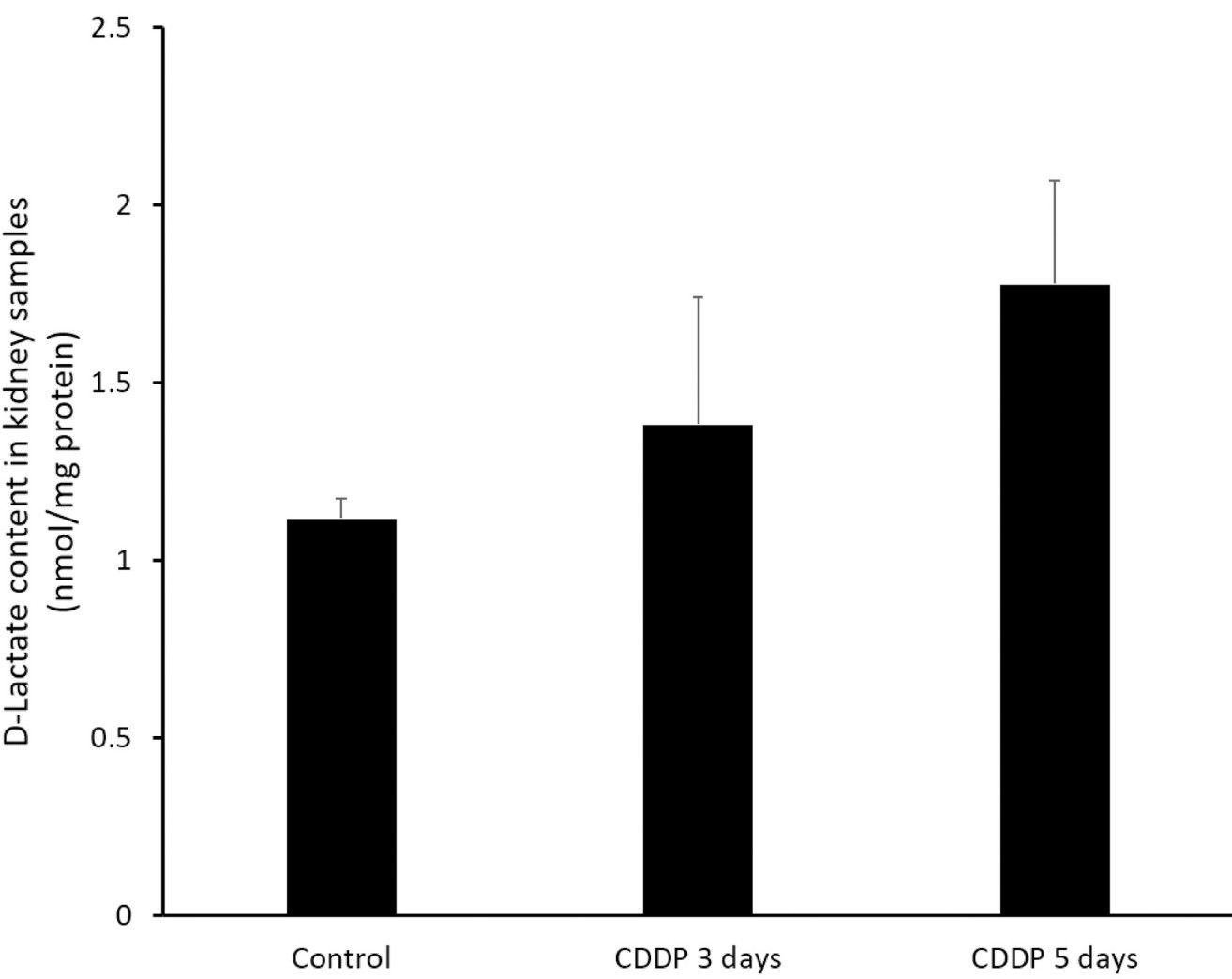

**Fig 5. Two-dimensional column-switching HPLC quantification of D-lactate in the kidney tissues.** The content of D-lactate was increased in CDDP 3-day and CDDP 5-day group, tended to increase as the MG content increased. N = 5 in each group, every experiment was conducted in triplicate.

### FD-LC-MS/MS proteomic analysis

The FD-LC-MS/MS chromatograms for the control, CDDP 3-day and CDDP 5-day groups are shown in Fig 6. The 33 and 9 peaks with significantly different peak heights in the CDDP 3-day and CDDP 5-day groups, respectively, compared to the control group were collected for LC-MS/MS analysis and identified using the MASCOT database.

The proteins identified in the CDDP 5-day group were mostly unremarkable, such as hemoglobin or cytoskeletal proteins (S2 Table). However, 11 antioxidant-related proteins were identified in the CDDP 3-day group (Fig 7; Table 3; S1 Table). From these 11 proteins, SOD-1 and DJ-1 were selected as target proteins and their expression levels were confirmed via western blotting.

### Western blotting

The semi-quantitative analysis of the western blots revealed DJ-1 and SOD-1 were expressed at 1.9-fold ($p < 0.05$) and 1.25-fold ($p < 0.05$) higher levels in the kidney tissue homogenates of

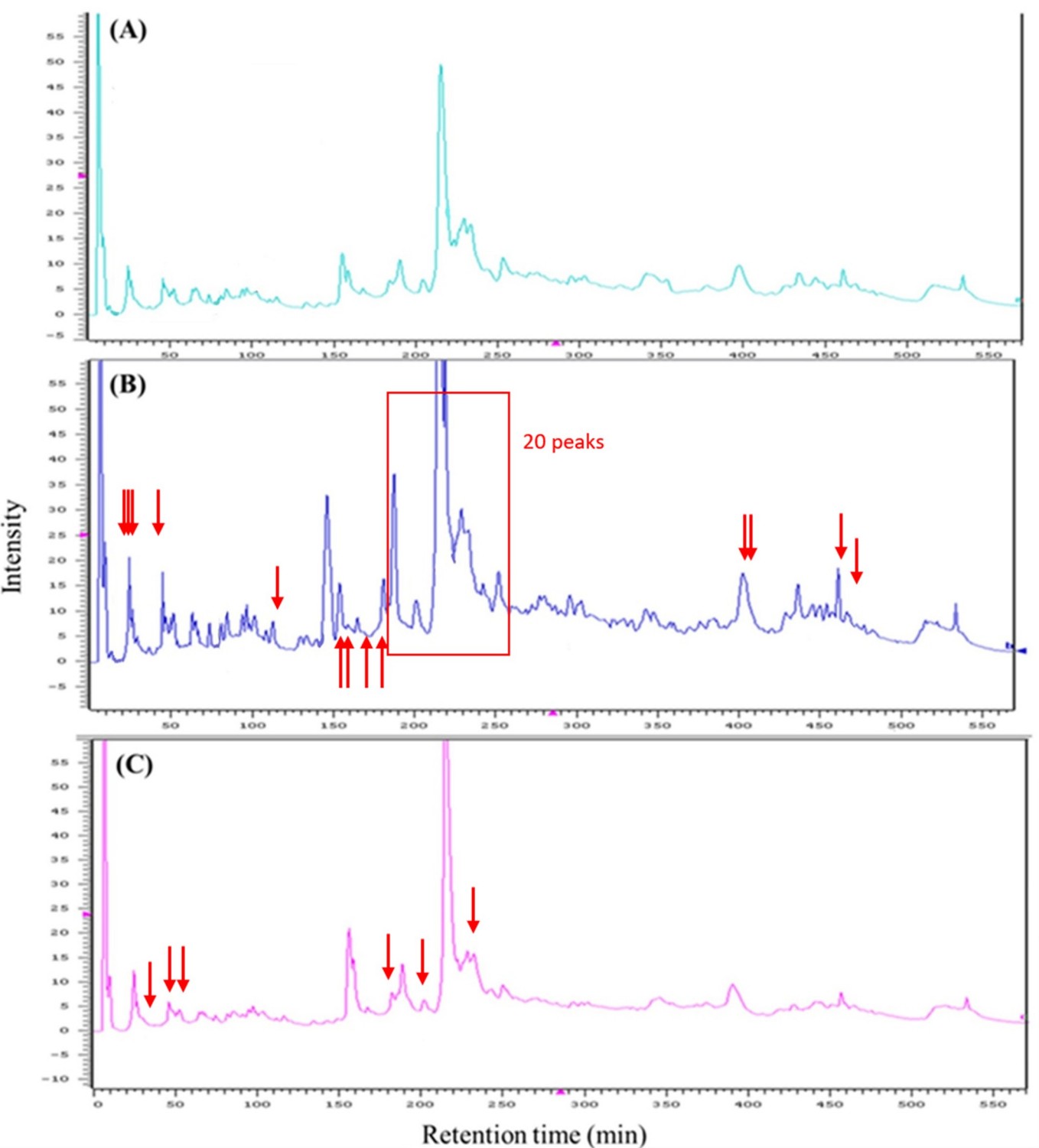

**Fig 6. FD-HPLC chromatograms of protein separation of the mouse kidney homogenates.** Compared to the control group (A), 33 peaks were significantly altered in the CDDP 3-day group (B) and 9 peaks were significantly altered in the CDDP 5-day group (C).

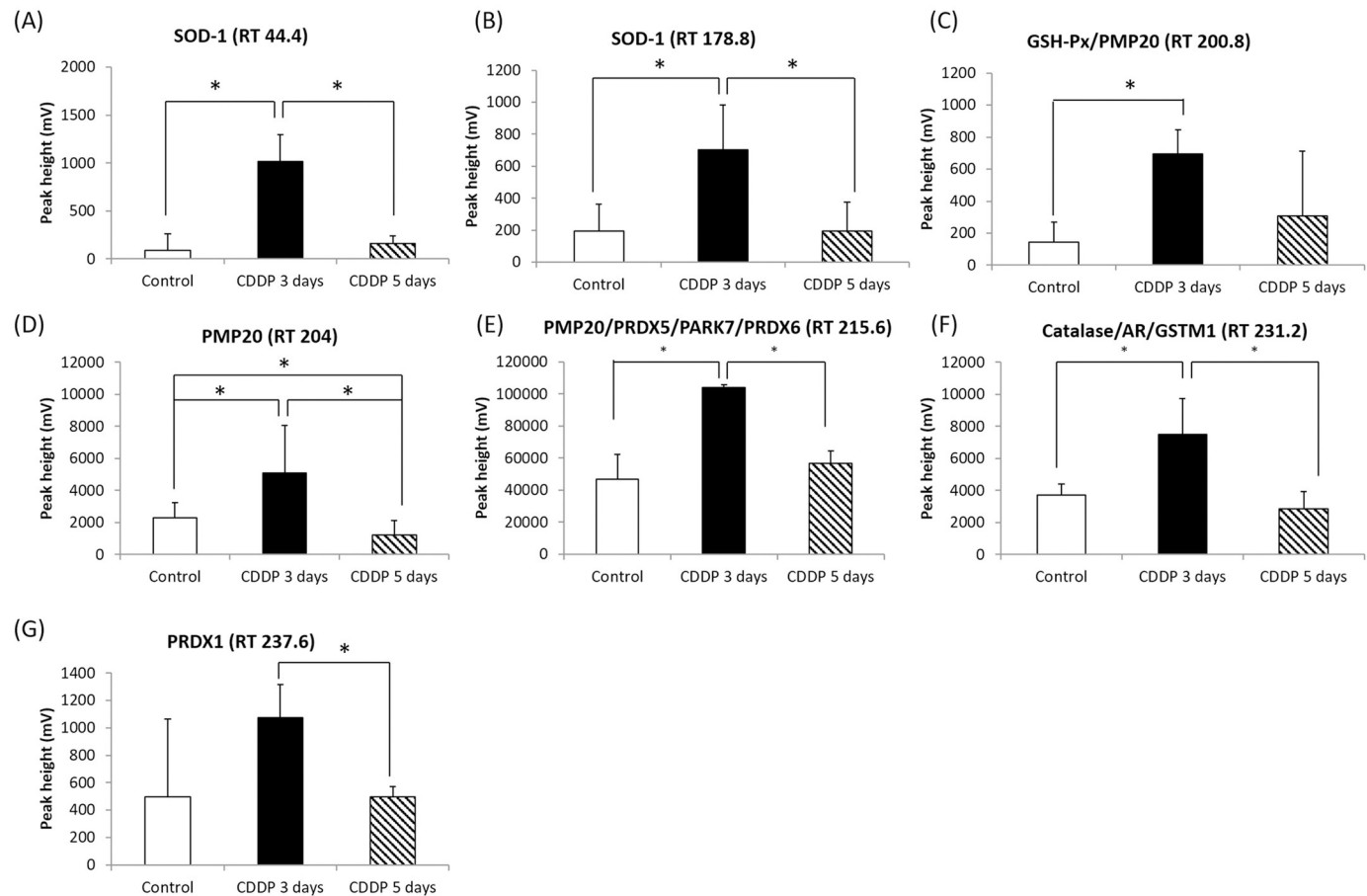

**Fig 7. FD-HPLC quantification of oxidative stress-associated proteins.** The oxidative stress-associated proteins in the kidney homogenates of the control group, CDDP 3-day group and CDDP 5-day group. The peaks were quantified based on peak height; *p < 0.05 vs. control group, Student's T-test.

**Table 3. The antioxidant-related proteins identified in kidney samples by FD-LC-MS/MS proteomic analysis using the MOSCOT database.**

| Retention time (min) | Proteins (Abbreviation) | MW. (kDa) | Score | GI NO. |
|---|---|---|---|---|
| 44.4 | Cu/Zn superoxide dismutase (SOD1) | 15,752 | 91 | gi\|226471 |
| 178.8 | Cu/Zn superoxide dismutase (SOD1) | 15,752 | 78 | gi\|226471 |
| 200.8 | Glutathione peroxidase (GSH-Px) | 22,276 | 55 | gi\|2673845 |
| 200.8 | Peroxisomal membrane protein 20 (PMP20) | 17,004 | 76 | gi\|6746357 |
| 204 | Peroxisomal membrane protein 20 (PMP20) | 17,004 | 76 | gi\|6746357 |
| 212.5 | Peroxisomal membrane protein 20 (PMP20) | 17,004 | 183 | gi\|6746357 |
| 215.6 | Peroxiredoxin V (PRDX 5) | 21,975 | 48 | gi\|6644338 |
| 215.6 | Parkinson disease protein 7 (DJ-1/PARK7) | 20,008 | 68 | gi\|55741460 |
| 218.5 | Peroxiredoxin-6 (PRDX6) | 24,855 | 64 | gi\|3219774 |
| 231.2 | Catalase | 36,564 | 90 | gi\|10946870 |
| 231.2 | Aldo-keto reductase family 1, member A4 | 46,555 | 83 | gi\|6996911 |
| 237.6 | Peroxiredoxin 1 (PRDX1) | 22,162 | 41 | gi\|6754976 |
| 231.2 | Glutathione S-transferase mu 1 (GSTM1) | 3,242 | 51 | gi\|50165 |

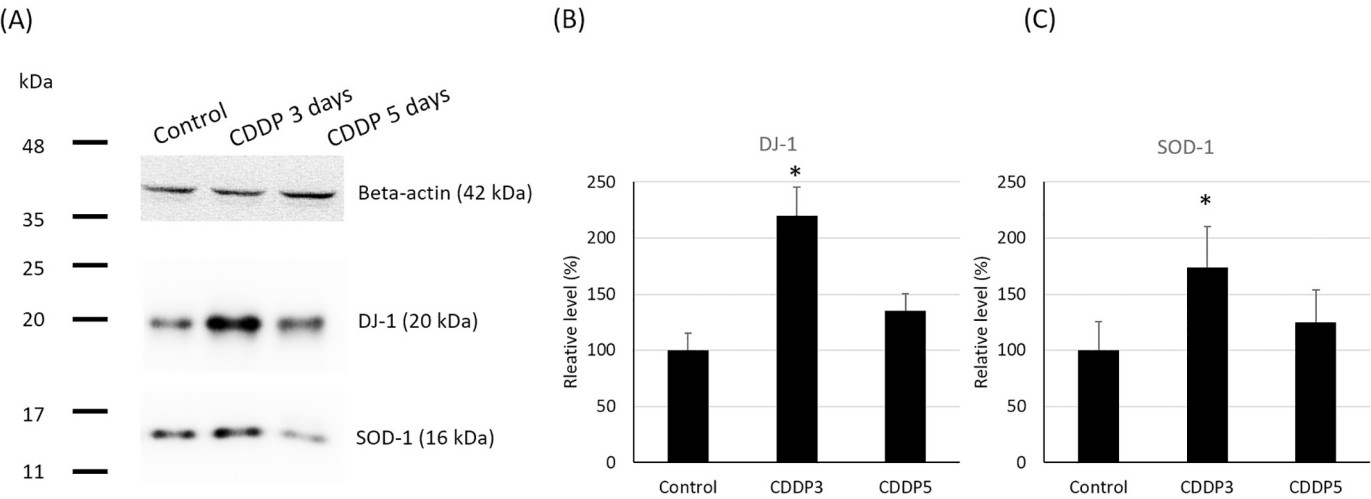

**Fig 8. Expression of DJ-1 and SOD1 in the kidney homogenate.** (A) Representative western blots of DJ-1, SOD-1 and β-actin expression. β-actin was used as the internal standard. (B-C) Quantification of DJ-1 and SOD-1 protein expression. SOD1: Cu-Zn superoxide dismutase; DJ-1/PARK7: Parkinson disease protein 7; $^*p < 0.05$ vs. control group, Student's T-test. N = 5 in each group, every experiment was conducted in triplicate.

the CDDP 3-day group than the control group (Fig 8). DJ-1 and SOD-1 also tended to be expressed at higher levels in the CDDP 5-day group than the control group, though the differences were not significant. These results were consistent with the FD-LC MS-MS proteomics study and confirmed that DJ-1 and SOD-1 were upregulated in the kidney after treatment with CDDP for 3 days.

## Discussion

We successfully induced acute renal injury in mice via intraperitoneal injection of CDDP for 3 or 5 days. The levels of urinary creatinine, BUN and NAG were significantly elevated in the CDDP 3-day and CDDP 5-day groups compared to the control group (Figs 1 and 2). Tubulointerstitial injury, such as tubular cell atrophy and cell infiltration [34, 35] were observed in the PAS-stained kidney sections of the mice treated with CDDP.

Development and progression of renal inflammation in response to CDDP is mediated via a complex network of inflammatory mediators. Among the numerous inflammatory cytokines, Volarevic *et al.* (2019) revealed that TNF-α is an important mediator of cisplatin-induced renal inflammation [5]. Moreover, CDDP-induced DNA damage upregulates TNF-α [6] and CDDP-induced AKI was attenuated in TNF-α-deficient mice or mice treated with TNF-α inhibitors or TNF-α neutralizing antibodies [36]. Renal parenchymal cells, macrophages and CD4[+] T helper lymphocytes have been shown to secrete TNF-α in CDDP-induced AKI [7].

ROS and oxidative stress may trigger increased production of TNF-α. The TNF-α signaling pathways that culminate in activation of NF-κB are influenced by ROS and lead to upregulation of antioxidant proteins, demonstrating that TNF-α and ROS influence each other via a positive feedback loop [37]. We found the levels of TNF-α were significantly elevated in the CDDP 3-day group and CDDP 5-day group. Taken together, these results indicate that CDDP increases oxidative stress, which may trigger increased production of TNF-α. These results suggest that TNF-α may be an important regulator of oxidative stress in CDDP-induced AKI.

The balance between the levels of ROS and antioxidant enzymes plays a crucial role in CDDP-induced AKI [1, 2]. MG contains two carbonyl groups and generates advanced

glycation end product adducts on proteins and nucleic acids, which lead to protein denaturation and malfunctions that can induce mitochondrial dysfunction and apoptosis [10–13]. In the present study, administration of CDDP for 3 and 5 days elevated the content of MG in the kidney (Fig 4). CDDP also tended to increase the levels of D-lactate (Fig 5), a metabolite of MG produced by the glyoxalase system [38]. Godbout *et al.* (2002) reported the presence of MG could enhance cisplatin-induced cytotoxicity: co-treatment of human myeloma cells with MG and cisplatin synergistically increased apoptosis (by 90% compared to the expected additive effect of MG and cisplatin) [39]. In addition, several studies have demonstrated MG may synergistically affect chemotherapy [40–42]. The levels of MG and its metabolite D-lactate correlated with oxidative stress and were associated with kidney injury in severe animal models [14, 19–21, 43]. Chou *et al.* (2015) demonstrated the levels of D-lactate correlated with the urinary albumin-to-creatinine ratio of diabetic nephropathy in humans [23]. In the present study, cisplatin increased the production of MG in the kidney, which was confirmed by the detection of elevated levels of the MG metabolite D-lactate. Moreover, the levels of MG and D-lactate were higher in mice administrated CDDP for 5 days than 3 days. Taken together, this study shows that MG may contribute to and represent an important pathological mechanism of CDDP-induced AKI.

We also conducted a proteomic study to identify proteins implicated in the association between MG and CDDP-induced nephrotoxicity. Two proteins were identified to be significantly associated with CDDP induced nephrotoxicity: SOD-1 and DJ-1. SOD-1 is a superoxide dismutase that localizes to the cytoplasm, peroxisomes and mitochondria. SOD-1 converts superoxide into hydrogen peroxide ($H_2O_2$), which is subsequently converted into water and oxygen by catalase or to water and oxidized glutathione by GSH-Px via consumption of glutathione disulfide [44]. Noori *et al.* (2010) demonstrated that CDDP-induced nephrotoxicity is related to oxidative stress, an unbalanced redox state, impairments to energy metabolism and increased apoptosis related to mitochondrial dysfunction in a rat model [45]. Our proteomic study and western blotting revealed that SOD-1 was significantly upregulated in the kidneys of the CDDP 3-day group (and tended to be higher in the CDDP 5-day group) than the control group. These results indicate CDDP increases the production of MG, which in turn leads to elevated oxidative stress. The increased expression of the ROS scavenging enzyme: SOD-1 in the kidneys of mice treated with CDDP may occur as a protective mechanism to compensate for increased oxidative stress [46].

The severe kidney damage in the CDDP 5-day group indicates the levels of oxidative stress exceeded the capacity of the tissues to compensate for oxidative stress, which would lead to a reduction in the levels of SOD-1. Moreover, high levels of protein denaturation and malfunctions induced by increased production of MG could also reduce the activity of anti-oxidative proteins in the kidneys of the CDDP 5-day group.

DJ-1 has been demonstrated to ameliorate oxidative stress by converting advanced glycation end products contributed by MG back to functional proteins and nucleotides [47–49]. Our proteomic study and western blotting showed the levels of DJ-1 were elevated in the CDDP 3-day group compared to the control group, indicating DJ-1 expression may increase as part of a mechanism to attenuate oxidative damage by converting advanced glycation end products back to functional proteins. However, in the CDDP 5-day group, the damage was beyond the self-repair capacity of the kidney and the cells had started to die, which in turn may decrease the expression of DJ-1. Interestingly, previous research reported the expression of c-Jun N-terminal kinases (JNKs) was mediated by TNF-α. The JNKs activate apoptosis, whereas DJ-1 inhibits apoptosis by inhibiting the expression of JNKs [50, 51]. Taken together, these results indicate DJ-1 may convert advanced glycation end products back to functional proteins and nucleotides to prevent cell death, and may also prevent apoptosis by inhibiting

JNKs as a mechanism to protect against oxidative stress. The mechanism by which CDDP increases the production of MG was not investigated in this study. Chen *et al.* (2020) reported inflammatory reaction might enhance glycolysis, leading the content of MG increment, which is one of the by-products from glycolysis [52]. Späth *et al.* conducted CDDP induced AKI animal model and investigate the altered protein by LC-MS proteomic method and concluded that endogenous fatty acid biosynthesis is a key natural protective mechanism to counteracted CDDP nephrotoxicity [53]. Allaman *et al* reported despite glycolysis, lipid metabolism is an endogenous MG source as well [54]. Which could support the MG source in this animal model is mainly endogenous. The exact mechanism still needs further rigorous investigation.

There are several studies focused on CDDP induced kidney damage using proteomic analyzing method, tried to explore the pathological mechanism of CDDP nephrotoxicity and find potential marker. While the model nor the analyte are not consisting [53, 55–59]. This study used mice as experimental animal and liquid chromatography-tandem mass spectrometry (FD-LC-MS/MS) proteomic analysis method and found DJ-1 and SOD-1 were two proteins could support the role of MG in CDDP induced AKI, which may could provide as a foundation for novel treatment or prevention strategy.

## Conclusions

We successfully established a CDDP-induced mouse model of AKI. Our results indicate CDDP induces oxidative stress and upregulates MG in the kidney. MG increases oxidative

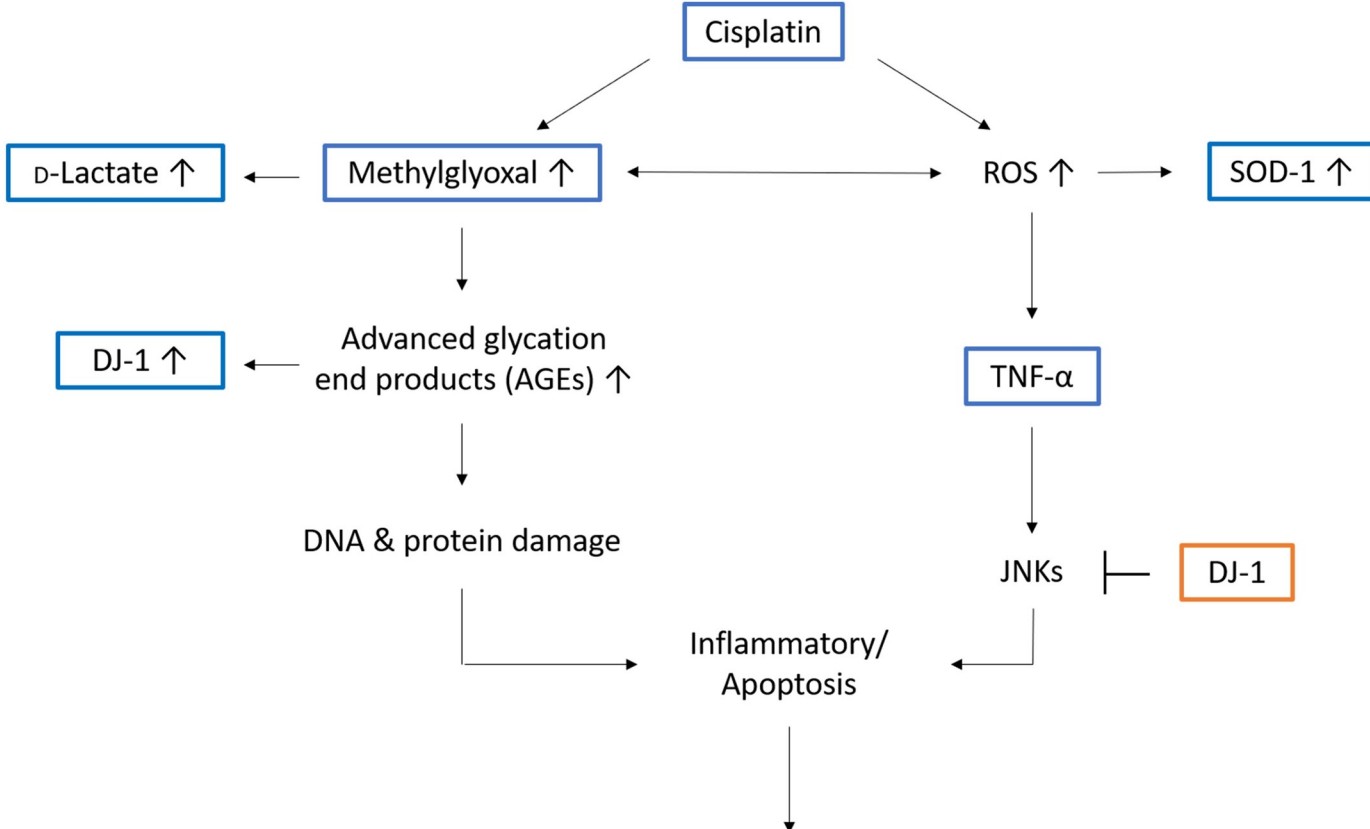

**Fig 9. Schematic illustration of the potential mechanism of cisplatin-induced acute kidney injury.** CDDP induces oxidative stress and upregulates MG in the kidney. MG increases oxidative stress and upregulate SOD-1. Higher DJ-1expression level could increase the conversion of AGE contributed by MG back to functional proteins and nucleotides to prevent cell death, inflammation reaction and apoptosis.

stress and may play a crucial role in the pathology of CDDP-induced kidney damage. Similarly, CDDP also increased the production of the MG metabolite D-lactate. Our FD-LC-MS/ MS proteomic study identified two antioxidant-related proteins, SOD-1 and DJ-1, were associated with CDDP-induced AKI. Based on our findings and the literature, we propose a mechanism for CDDP-induced AKI (Fig 9). The mechanism by which CDDP increases the production of MG was not investigated in this study and merits further investigation.

## Supporting information

**S1 Fig. FD-HPLC chromatograms for the derivatives of methylglyoxal (MG) in the kidney tissues of mice in the control group, CDDP 3-day group and CDDP 5-day group.**
(PDF)

**S2 Fig. FD-HPLC chromatogram for the derivatives of D-lactate in the kidney tissues of the mice in the control group, CDDP 3-day group and CDDP 5-day group.**
(PDF)

**S1 Table. Differential proteins identified in the kidney tissues of the CDDP 3-day group vs. control mice.**
(PDF)

**S2 Table. Differential proteins identified in the kidney tissues of the CDDP 5-day group vs. control mice.**
(PDF)

**S1 Raw Images.**
(TIF)

**S2 Raw Images.**
(TIF)

**S3 Raw Images.**
(TIF)

**S1 File.**
(PDF)

## Acknowledgments

We are grateful to the Yung Shin Pharmaceutical Co. (Taiwan) for providing an API 4000 triple quadrupole mass spectrometer. We appreciate Professor Shiro Ueda shared his design of the animal metabolic cages to our team.

## Author Contributions

**Conceptualization:** Jen-Ai Lee.

**Formal analysis:** Pei-Yun Tsai.

**Funding acquisition:** Shih-Ming Chen.

**Methodology:** Shih-Ming Chen, Jen-Ai Lee.

**Project administration:** Tsung-Hui Chen, Hui-Ting Chang, Tzu-Yao Lin, Chia-Yu Lin, Pei-Yun Tsai.

**Resources:** Chien-Ming Chen.

**Software:** Chien-Ming Chen.

**Supervision:** Kazuhiro Imai, Chien-Ming Chen.

**Writing – original draft:** Tsung-Hui Chen.

**Writing – review & editing:** Tzu-Yao Lin, Chia-Yu Lin, Kazuhiro Imai, Jen-Ai Lee.

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
