## [Decision Letter · Decision Letter 0]

11 May 2020

PONE-D-20-05819

Methylglyoxal and D-Lactate in Cisplatin-Induced Acute Kidney Injury: Investigation of the Potential Mechanism via Fluorogenic Derivatization Liquid Chromatography-Tandem Mass Spectrometry (FD-LC-MS/MS) Proteomic Analysis

PLOS ONE

Dear Prof. Lee,

Thank you for submitting your manuscript to PLOS ONE. After careful consideration, we feel that it has merit but does not fully meet PLOS ONE’s publication criteria as it currently stands. Therefore, we invite you to submit a revised version of the manuscript that addresses the points raised during the review process.

To consider  further the paper for publication, all the issue raised by the reviewers must be accurately addressed. In particular, appropriate controls should be included in figure 3 and the entire and coherent  western blot image should be  showed in figure 8.

We would appreciate receiving your revised manuscript by Jun 25 2020 11:59PM. To enhance the reproducibility of your results, we recommend that if applicable you deposit your laboratory protocols in protocols.io, where a protocol can be assigned its own identifier (DOI) such that it can be cited independently in the future. For instructions see: http://journals.plos.org/plosone/s/submission-guidelines#loc-laboratory-protocols

We look forward to receiving your revised manuscript.

Kind regards,

Fabio Sallustio

Academic Editor

PLOS ONE

Journal Requirements:

2. Thank you for stating in the manuscript: 'All animal protocols followed the ethical guidelines of the Institutional Animal Care and Use Committee of Taipei Medical University (LAC-2019-0167).

a. Please amend your current ethics statement to confirm that your named ethics committee specifically approved this study.

For additional information about PLOS ONE submissions requirements for ethics oversight of animal work, please refer to http://journals.plos.org/plosone/s/submission-guidelines#loc-animal-research

Reviewers' comments:

Reviewer's Responses to Questions

**Comments to the Author**

1. Is the manuscript technically sound, and do the data support the conclusions?

Reviewer #1: Partly

Reviewer #2: No

2. Has the statistical analysis been performed appropriately and rigorously? 

Reviewer #1: Yes

Reviewer #2: No

3. Have the authors made all data underlying the findings in their manuscript fully available?

Reviewer #1: Yes

Reviewer #2: No

4. Is the manuscript presented in an intelligible fashion and written in standard English?

Reviewer #1: Yes

Reviewer #2: No

5. Review Comments to the Author

Reviewer #1: Overall, this is an interesting manuscript. The rationale and literature review are appropriate. In general, the data support the conclusions reached by the authors. However, there are several points that need to be addressed by the authors. There are a couple of general comments followed by a more specific listing. First, in Figure 1-4 and 5-8, the text and figure caption mention that the comparisons are performed versus a control group. The figures mention a "normal" group. The labeling needs to be consistent with the text of the manuscript. I recommend changing the label on the figures to 'control'. The other general comment has to do with Tables 1 and 2. Table 1 shows a 9.5 hour gradient for FD-HPLC and Table 2 show a 45 min gradient for proteomics analysis. Yet Figure 6, shows chromatograms for the protein separation and the text states "FD-LC-MS/MS chromatograms". Which gradient was used as the time scales are not consistent with the experimental section?

Specific comments:

1. Page 5, line 7: "PAGE" needs to be defined.

2. Page 7, lines 2-5: The authors should state that the additional MS and MASCOT details are provided later in the manuscript.

3. Page 7, lies 10-12: What are the four animal groups? How many animals per group? need additional detail here.

4. Page 15, line 19-20: What are the search criteria used?

5. Page 19, Figure 1: What is 'n' for each group? This information should be provided all the figures and in the figure captions.

6. Page 26, Figure 4 caption: This caption is inadequate. Additional information in needed.

7. Page 29, Figure 7: Again, the retention times of the identified proteins do not match the gradient given in Table 3.

After the authors have addressed these issues, the manuscript should be in a form suitable for publication.

Reviewer #2: This paper by Chen et al. analyzes metabolites and proteins in cisplatin kidney injury model.

The following points deserve the authors’ attention:

1. The Figure 2 lacks scale bars and the different groups have different magnification. Not sure if anything can be concluded from them.

2. The data in Figure 3 lacks positive and negative control and does not corroborate previous report son the role of TNFalpha in this model. The according literature is also not mentioned.

3. The data in Figure 6 is not very clear and needs better representation with clear mass spectra, since derivates are depicted.

4. The immunblot data in FIguer 8 does not represent the data shown in the quantification. Also MW markers should be added.

5. Methods: From the methods section regarding the fluorescent derivatization, the experiments cannot be reproduced. It is not clear how proteins were identified since no tryptic digestion step is performed. If this was top down, more information on peptide/protein identification is needed. Like this, bioinformatic analysis is not feasible.

6. Discussion:Very recent cisplatin proteomics studies have been performed. However, these studies are not mentioned and should be used to compare with the authors’ proteomics findings. The following studies were not mentioned: PMID: 25450742, PMID: 30522767, PMID: 31493026, PMID: 31629959, PMID: 17021608PMID: 24265863

6. PLOS authors have the option to publish the peer review history of their article (what does this mean?). If published, this will include your full peer review and any attached files.

Reviewer #1: No

Reviewer #2: No

---

## [Author Response · Author response to Decision Letter 0]

11 Jun 2020

First of all, we appreciated both reviewers to point out the blind points that we did not notice or should state more clearly in the beginning. Here are our responses for the specific comments by each reviewer.

Response to general comment by reviewer 1:

 We appreciate for you kindly remind us the writing mistake. Figure captions had been revised according to your suggestion. 

To response the question by regarding to the retention time of protein separation chromatograms (figure 6). Table 1 shows a 570 min gradient for the HPLC protein separation of mice kidney homogenate. Then, the separated sample were analyzed by LC-MS/MS, which using a 45 mins gradient which shows in Table 2. Figure 6. indicate the result of HPLC protein separation, which the scale does correspond with the 570 min gradient program. The retention time in Table 3 is the MSSCOT database comparison results and the peak height in Figure 7 indicate the protein expression in each group. Here we would like to apologize for the low resolution of the picture of chromatogram due to the software issue. We had tried our best to improve the quality of the pictures.

Response to specific comment 1: The abbreviation had been changed into “Two-dimensional polyacrylamide gel electrophoresis (2D-PAGE)”. 

Response to specific comment 2: The MASCOT research detail had been added into the revised draft, please see the attached file "Response to reviewers".

Response to specific comment 3:The sections describe animal grouping had been revised, please see the attached file "Response to reviewers".

Response to specific comment 4: The MASCOT research detail had been added into the revised draft, please see the attached file "Response to reviewers".

Response to specific comment 5: The figure captions had been revised, please see the attached file "Response to reviewers".

Response to specific comment 6: The figure caption had been revised, please see the attached file "Response to reviewers".

Response to specific comment 7: Regarding to the retention time of chromatograms. Our answer is in the response to general comments. In brief, Table 1 shows a 570 min gradient program and Figure 6. indicate the result of HPLC protein separation. The retention time in Table 3 is the MSSCOT database comparison results and the peak height in Figure 7 indicate the protein expression in each group.

Response to comments by reviewer 2: 

Response to comment 1: To add the scale bar to Figure 2, we had obtained a new set of histological sections’ pictures, and rescored the section. The result consists with pervious scoring result. Scale bar had added to the revised figure.

Response to comment 2: 

a) Regarding to the positive or negative control in the immunofluorescence image, we do recognize the importance of the controls, which could provide a clearer and solid evidence to enhance the causality. In this research, immunofluorescence analysis was conducted on the section of mice kidney samples, the negative control could achieve by using TNF-α gene knock-out mice. Considering the weight of benefit using gene mice and to reduce using experimental animals, we did not include negative control into our experiment design. Furthermore, in our previous published article, the immunofluorescence analysis conducted in mice kidney sections did not include negative control, neither (PMID: 31968011). Therefore, we believe without negative control could still represent the result correctly.

b) Our immunofluorescence analysis result of TNF-α revealed the level of TNF-α was correlated with the kidney damage by observing the biomedical analysis and histological images. ROS attributed by methylglyoxal is a key concept in our study (Figure 9). We believed CDDP could increase the level of methylglyoxal and elevate ROS, which trigger the increased production of TNF-α.

Response to comment 3: As the response to Reviewer #1, we apologize for the low resolution of chromatogram due to the software issue. We had tried our best to improve the quality of the pictures and the figure had been revised to improve resolution.

Response to comment 4: To add the molecule weight marker to the western blot image, we re-conducted gel electrophoresis and semi-quantification. The trend consisted with the previous results. The images had been revised.

Response to comment 5: Our protein separation method included trypsin digestion step. After the separation by HPLC, 2.5 μL of 10 mM CaCl2, 20 μL of 50 mM NH4CO3 and 2.5 μL trypsin were added to the protein residues and incubated at 37 °C for 2 h for the digestion. And this protein identification method had been published in several articles (PMID: 31968011, PMID: 19714884, PMID: 29088495, PMID: 22972526), we have confidence that this method is dependable. And the detail MASCOT research had been revised, please see the attached file "Response to reviewers".

Response to comment 6: Here, we are grateful for the reviewer raising the point that we did not attention previously. After we read the references that Reviewer #2 provided, the discussion in the main draft was revised as below, and references provided by the reviewer had been cited. The discussion in main draft had been revised, please see the attached file "Response to reviewers".

---

## [Decision Letter · Decision Letter 1]

19 Jun 2020

PONE-D-20-05819R1

Methylglyoxal and D-Lactate in Cisplatin-Induced Acute Kidney Injury: Investigation of the Potential Mechanism via Fluorogenic Derivatization Liquid Chromatography-Tandem Mass Spectrometry (FD-LC-MS/MS) Proteomic Analysis

PLOS ONE

Dear Dr. Lee,

Thank you for submitting your manuscript to PLOS ONE. After careful consideration, we feel that it has merit but does not fully meet PLOS ONE’s publication criteria as it currently stands. Therefore, we invite you to submit a revised version of the manuscript that addresses the minor points raised during the review process.

We look forward to receiving your revised manuscript.

Kind regards,

Fabio Sallustio

Academic Editor

PLOS ONE

Reviewers' comments:

Reviewer's Responses to Questions

**Comments to the Author**

1. If the authors have adequately addressed your comments raised in a previous round of review and you feel that this manuscript is now acceptable for publication, you may indicate that here to bypass the “Comments to the Author” section, enter your conflict of interest statement in the “Confidential to Editor” section, and submit your "Accept" recommendation.

Reviewer #1: All comments have been addressed

2. Is the manuscript technically sound, and do the data support the conclusions?

Reviewer #1: Yes

3. Has the statistical analysis been performed appropriately and rigorously? 

Reviewer #1: Yes

4. Have the authors made all data underlying the findings in their manuscript fully available?

Reviewer #1: Yes

5. Is the manuscript presented in an intelligible fashion and written in standard English?

Reviewer #1: Yes

6. Review Comments to the Author

Reviewer #1: The authors do an adequate job of addressing previous reviewer concerns. However, there are several minor points/wording issues that the authors should address. These points are listed below in order of appearance in the manuscript.

1. Page 7, lines 13-14: The sentence should be written as: "Five animals were allocated into each group."

2. Page 16, lines 1-5: The paragraph need to be rewritten. Suggestion "After detection, data was submitted to MASCOT, a protein identification program .... MASOCT is widely-used protein identification search engine. Line 5: the word "collate" should be deleted.

Figure captions 1, 2, 3, 4, 5 and 8: The words "triple repeat" should be deleted and replaced with 'in triplicate'

7. PLOS authors have the option to publish the peer review history of their article (what does this mean?). If published, this will include your full peer review and any attached files.

Reviewer #1: No

---

## [Author Response · Author response to Decision Letter 1]

21 Jun 2020

This letter is to response to the issues raised by Reviewer in minor revision.

Again, we are appreciated to Reviewer’s kindly remind us the writing and grammarly mistakes. We had amended the paragraph according to the Reviewer’s suggestion. The following is the response to the comments from Reviewer.

Comment 1: Page 7, lines 13-14: The sentence should be written as: "Five animals were allocated into each group."

Response: The paragraph had been revised, please see the attached file "Response to reviewer".

Comment 2: Page 16, lines 1-5: The paragraph need to be rewritten. Suggestion "After detection, data was submitted to MASCOT, a protein identification program .... MASOCT is widely-used protein identification search engine. Line 5: the word "collate" should be deleted.

Response: The paragraph had been revised, please see the attached file "Response to reviewer".

Comment 3: Figure captions 1, 2, 3, 4, 5 and 8: The words "triple repeat" should be deleted and replaced with 'in triplicate'

Response: The figure caption had been revised, please see the attached file "Response to reviewer".

---

## [Editor Report · Decision Letter 2]

24 Jun 2020

Methylglyoxal and D-Lactate in Cisplatin-Induced Acute Kidney Injury: Investigation of the Potential Mechanism via Fluorogenic Derivatization Liquid Chromatography-Tandem Mass Spectrometry (FD-LC-MS/MS) Proteomic Analysis

PONE-D-20-05819R2

Dear Dr. Lee,

We’re pleased to inform you that your manuscript has been judged scientifically suitable for publication and will be formally accepted for publication once it meets all outstanding technical requirements.

Kind regards,

Fabio Sallustio

Academic Editor

PLOS ONE
---

## [Editor Report · Acceptance letter]

30 Jun 2020

PONE-D-20-05819R2 

Methylglyoxal and D-lactate in cisplatin-induced acute kidney injury: Investigation of the potential mechanism via fluorogenic derivatization liquid chromatography-tandem mass spectrometry (FD-LC-MS/MS) proteomic analysis 

Dear Dr. Lee:

I'm pleased to inform you that your manuscript has been deemed suitable for publication in PLOS ONE. Congratulations! Your manuscript is now with our production department. 

Kind regards, 

on behalf of

Dr. Fabio Sallustio 

Academic Editor

PLOS ONE